# OpenReview forum: "GraphPrompt: Black-box Jailbreaks via Adversarial Visual Knowledge Graphs"
_ICLR.cc/2026/Conference — ICLR 2026 Conference Withdrawn Submission_

### Official Review · Reviewer_1Aot · 2025-10-29

**Soundness:** 3
**Presentation:** 2
**Contribution:** 2
**Rating:** 4
**Confidence:** 5

**Summary:**

This paper introduces GraphPrompt, a typographic jailbreaking attack on Multimodal Large Language Models (MLLMs) that utilizes a Visual Knowledge Graph (VKG). To break the alignment of the MLLM, GraphPrompt embeds a jailbreak prompt into a VKG. It begins by constructing a Knowledge Graph (KG) using a Large Language Model (LLM), which is then transformed into a VKG using Mermaid. The VKG is subsequently input into the target MLLM through the visual channel alongside a benign textual prompt. The effectiveness of the jailbreaking is evaluated using an LLM-based judge model. Experiments demonstrate that GraphPrompt successfully jailbreaks various MLLMs with a high Attacking Success Rate (ASR), underscoring the vulnerabilities of MLLMs in processing VKGs.

**Strengths:**

1. This paper is well organized and easy to follow.
2. Embedding malicious intention into VKG appears to be novel.
3. Experiments show GraphPrompt is promising for jailbreaking SOTA MLLM.

**Weaknesses:**

1. MLLMs are known to be vulnerable to typographic jailbreaking attacks, where malicious textual questions are converted into images. This approach takes advantage of the model's image understanding capabilities to circumvent textual filters, effectively "breaking the safety alignment." Consequently, the novelty of transforming harmful textual knowledge graphs into typographic images (VKGs) is somewhat limited. While GraphPrompt shows better performance than FigStep, the underlying reasons for this difference have not been thoroughly examined.

2. There is limited discussion on defense methods against GraphPrompt. Although the Related Work section includes a paragraph on defenses against jailbreaking, there are no experiments investigating how basic OCR-based filters or input-moderation defenses (e.g., converting the images into a textual description as a supplement to the benign textual input) can reduce the effectiveness of GraphPrompt.

**Questions:**

Please refer to the Weaknesses part.

---

> ### Author Response · Authors · 2025-11-20
>
> We thank Reviewer 1Aot for the careful evaluation of our work and for the positive comments on the paper’s organization, the novelty of embedding malicious intent into VKGs, and the empirical strength of GraphPrompt against SOTA MLLMs. Below we outline how we plan to respond to the two main concerns about (i) novelty beyond existing typographic attacks, and (ii) the scope of our defense evaluation.
>
> ### **W1: Novelty beyond typographic jailbreaking (GraphPrompt vs. FigStep)**
>
> >MLLMs are already known to be vulnerable to typographic jailbreaking (e.g., turning harmful text into images). From this perspective, transforming harmful textual KGs into VKGs may seem like a limited extension. While GraphPrompt outperforms FigStep, the underlying reasons are not thoroughly examined.
>
>
> We appreciate this concern and agree that GraphPrompt must be clearly distinguished from prior typographic attacks such as FigStep. Our method is **not** simply “rendering harmful text as an image.” Instead, GraphPrompt uses a **three-layer obfuscation pipeline** (Figure 1):
>
> 1. the harmful query is first rewritten into a benign-looking, high-level description;
> 2. the harmful intent is then **decomposed and distributed** across different nodes and topological relations in a VKG;
> 3. this VKG is visually rendered and paired with a benign user prompt so that the model perceives it as a normal structured task.
>
> In other words, the core malicious intent is no longer present as an explicit sentence (as in typical typographic attacks), but is **encoded into the graph structure and node semantics**. This structural embedding and three-layer obfuscation are key to GraphPrompt’s high attack success rate.
>
> Beyond empirical ASR differences, we also investigate *why* GraphPrompt outperforms FigStep in Sec. 4.4 and Appendix E. **We apply the HiddenDetect methodology to Qwen-VL-Chat** to analyze why GraphPrompt achieves higher attack success rates than FigStep. For each transformer layer, we measure how strongly hidden states align with a learned “refusal direction.” The results show that:
>
> * text-only harmful prompts most strongly activate safety-critical layers;
> * FigStep and MM-SafetyBench reduce this activation;
> * **GraphPrompt’s VKGs produce the lowest refusal strength in these safety layers**, indicating that the model’s internal safety circuitry is even more suppressed for VKGs than for figstep attacks.
>
> These findings support our claim that GraphPrompt exploits a **structured-visual “parse-then-execute” pathway** that is qualitatively different from OCR-style typographic jailbreaks.

---

> ### Author Response · Authors · 2025-11-20
>
> ### **W2: Defense methods against GraphPrompt**
>
> >_Defense discussion is limited. While related work mentions jailbreak defenses, the paper lacks experiments with simple defenses such as OCR-based filters or input moderation (e.g., converting images back to text and feeding it to the safety layer)._
>
> Thank you for raising this important point. We agree that understanding how to mitigate VKG-based attacks is crucial, and we have made this aspect more explicit in the revised manuscript (Sec. 4.5 and Appendix C).
>
> Concretely, we implement and evaluate a system-level, prompt-based defense. As described in Sec. 4.5 Appendix C, we prepend an _intent-first safety system prompt_ (Figure 8) to the target MLLM. This prompt requires the model to:
> 1. first **summarize the intent of the VKG image in natural language**,
> 2. then explicitly **decide whether this intent violates safety policies**, and
> 3. only provide a detailed answer if it judges the intent to be safe.
>
> In other words, the model is explicitly asked to understand the image’s intent and perform a safety check _before_ continuing along the usual parse–then–execute pathway.
>
> We evaluate this defense on the same GraphPrompt-generated VKGs across all six target MLLMs. As reported in Appendix C (Table 5), this intent-first prompt **reduces ASR for every model**, but **does not eliminate** VKG-based jailbreaks: GraphPrompt still achieves non-trivial ASR under this stricter setting. This shows that “converting” the image into an internal textual description and asking the model to self-moderate is only _partially effective_ and does not fully close the structured-visual attack surface.
>
> We therefore explicitly point out in Sec. 6 that more powerful defenses are needed for VKG-style jailbreaks.
>
>
> **Table 5:** Effect of adding a system-level safety defender on attack success rate (ASR, %).
> Each entry is computed over 20 harmful queries per model.
> The last row reports ΔASR (defender − no defender, in percentage points).
>
> | Prompt type            | GPT-4o | GPT-5mini | GPT-5 | Qwen 2.5 | Claude | Gemini |
> |------------------------|:------:|:---------:|:-----:|:--------:|:------:|:------:|
> | No defender (ASR)      |   90   |    85     |  95   |   100    |   80   |  100   |
> | +System defender (ASR) |   65   |    75     |  70   |    95    |   60   |   95   |
> | ΔASR (pp)              |  -25   |    -10    | -25   |    -5    |  -20   |   -5   |

---

> ### Author Response · Authors · 2025-11-20
>
> Dear Reviewer 1Aot,
>
> Again, we appreciate your careful review and constructive comments. We hope that our clarifications on how GraphPrompt goes beyond typographic jailbreaks through structured-visual, topology-based obfuscation, our mechanistic analysis contrasting VKGs with FigStep using HiddenDetect, and our expanded experimental defense evaluation with the intent-first system prompt help address your concerns.
>
> Best regards,
>
> The Authors

---

### Official Review · Reviewer_JRvZ · 2025-10-30

**Soundness:** 3
**Presentation:** 3
**Contribution:** 2
**Rating:** 4
**Confidence:** 3

**Summary:**

This paper introduces GraphPrompt, a novel black-box jailbreaking attack framework for Multimodal Large Language Models (MLLMs). The core idea is to encode harmful intent not in plain text, but within the topological structure and visual cues of Visual Knowledge Graphs (VKGs). The attack pairs these adversarial VKG images with a benign-looking textual prompt (e.g., "analyze the tasks in this graph"), which tricks MLLMs into bypassing text-based safety filters and executing the embedded harmful instruction through a "covert parsing-execution pathway".

**Strengths:**

1. **High Efficacy and Realistic Threat Model**: The most significant strength is the attack's high effectiveness. Achieving a 94.0% average ASR—and rates as high as 100% on Gemini and 98% on GPT-5 and Qwen2.5-VL—is impressive. This is accomplished under a strict black-box assumption (no access to weights or gradients), making it a practical and realistic threat.

2. **Data Generation Pipeline**: The framework's ability to "automatically construct high-quality adversarial sample datasets" is a useful contribution. This provides a scalable method for red-teaming MLLMs against this new structured-visual threat dimension.

3. **Ablation Studies**: The ablation studies provide a clear picture of why the attack works. The findings that topology and resolution are the dominant factors, while visual elements like color and background are "second-order", are key takeaways that can inform future defense strategies.

**Weaknesses:**

1. **Limited and Potentially Insufficient Evaluation Dataset**: The primary weakness is the reliance on the SafeBench-Tiny dataset, which contains only 50 harmful queries. While the authors justify this for "reproducibility and experimental control", claiming a 94-100% ASR based on such a small sample size is a major overstatement. The high success rates could be an artifact of these specific 50 queries, and the results may not generalize to a more diverse and larger-scale benchmark.

2. **Questionable Novelty Compared to Prior Work (FigStep)**: The paper claims to be the first to leverage VKGs , but its novelty relative to existing "text-in-image" attacks like FigStep  is not sufficiently established. FigStep also works by decomposing instructions into steps and rendering them in an image. While the mechanism is interesting, the motivation for jailbreaking MLLMs is a very crowded research area. The paper does not sufficiently motivate why this specific vector is substantially different or more dangerous than the "abundant llm jailbreak attacks" that already exist such as FigStep.

3. **Lack of Experimental Defense Evaluation**: The paper discusses potential defenses in the conclusion, such as "structure-aware safety filtering" and "uncertainty-aware refusal". However, it presents no experiments to evaluate the efficacy of these or any other defenses. An attack paper is made much stronger by demonstrating how the uncovered vulnerability might be patched.

**Questions:**

See Weaknesses

---

> ### Author Response · Authors · 2025-11-20
>
> We thank Reviewer JRvZ for the detailed and thoughtful evaluation of our work, and we appreciate your positive comments on the high ASR under a strict black-box threat model, the usefulness of our data-generation pipeline, and the informativeness of our ablation studies. Below we outline how we respond to your main concerns about (i) dataset scale, (ii) novelty relative to FigStep and other jailbreaks, and (iii) experimental defense evaluation.
>
>
> ### **W1: Limited / potentially insufficient evaluation dataset**
>
> >_Relying on SafeBench-Tiny (50 harmful queries) makes a 94–100% ASR claim potentially overstated; the results may not generalize beyond this small set._
>
> We agree that 50 harmful queries is a relatively small evaluation set. Our goal is not to claim that _every_ harmful query will see 94–100% ASR, but to argue that the **observed advantage of GraphPrompt is robust rather than an artifact of this particular subset**. We support this from two angles:
>
> 1. **Reliability of the empirical effect.**
>     Even on this modest dataset, the effect sizes are strong and consistent:
>
>     - GraphPrompt **substantially outperforms** MM-SafetyBench and FigStep on **all six MLLMs**, not just a subset of models.
>
>     - The same pattern holds **across harmful-query categories** and in our ablation studies: changes in topology and resolution systematically affect ASR in the same direction for all models, while color/background remain second-order.
>
>     - Our efficiency and refusal metrics (fewer attempts, higher first-try success, near-zero explicit refusals) are aligned with the ASR gains, not in tension with them.
>
>
>     Together, these results suggest a **systematic behavior** of GraphPrompt across models and categories, rather than a few “easy” queries inflating the average.
>
> 2. **Mechanistic evidence that goes beyond this specific dataset.**
>     In addition, we conduct a **HiddenDetect-based analysis on Qwen-VL-Chat** (Sec. 4.4 and Appendix E) to probe _why_ GraphPrompt is more effective than text-only and typographic attacks:
>     - Text-only harmful inputs strongly activate safety-critical layers.
>     - FigStep / MM-SafetyBench images reduce this activation.
>
>     - GraphPrompt VKGs show the **lowest alignment with refusal directions** in these layers, indicating that the model’s internal safety signal is  suppressed when it processes VKGs as structured workflows.
>
>     This analysis depends on **internal representations and safety layers**, not on any single SafeBench-Tiny query, and thus supports the view that GraphPrompt is exploiting a **model-level vulnerability in the structured-visual “parse-then-execute” pathway**, rather than overfitting to a small set of prompts.
>
> Taken together, these two aspects — the consistent, large empirical gains across six models and metrics, and the mechanistic evidence that VKGs weaken safety-layer activation — strongly support that our findings are not artifacts of SafeBench-Tiny and that GraphPrompt’s behavior can reasonably be expected to generalize to larger and more diverse harmful-query sets, even though the exact ASR values may vary across datasets.

---

> ### Author Response · Authors · 2025-11-20
>
> ### **W2: Novelty compared to FigStep and other jailbreaks**
> >_The motivation space for jailbreak attacks is crowded, and it is unclear why VKG-based GraphPrompt is substantially different or more dangerous than existing “text-in-image” attacks like FigStep, which also decompose instructions into steps and render them visually._
>
> Our method is **not** just rendering harmful text as an image, but using a **three-layer obfuscation pipeline** tailored to VKGs:
>
> 1. **Role-play rewriting**: the harmful query is first rewritten into a benign-looking description (e.g., lecture, analysis), hiding explicit intent.
> 2. **Topology-borne encoding**: the harmful procedure is **decomposed and spread across nodes and edges** of a VKG (entities, stages, control flow), so the intent lives in the graph’s structure rather than a single dangerous sentence.
> 3. **VKG rendering + benign prompt**: the VKG is rendered and paired with a normal prompt like “analyze the workflow in this graph,” steering the model into a **parse-then-execute** reasoning mode.
>
>
> This is a different attack surface from FigStep, which mainly relies on readable text-in-image. Empirically, GraphPrompt consistently achieves **much higher ASR than FigStep on all six MLLMs** under the same harmful queries. Mechanistically, our HiddenDetect-based analysis on Qwen-VL-Chat (Sec. 4.4, Appendix E) shows that VKGs **suppress safety-layer “refusal” activations more strongly** than both text-only and FigStep inputs, indicating that the model is internally treating VKGs as benign structured tasks.
>
> Together, the **three-layer obfuscation** and the observed **internal safety-layer suppression** highlight that GraphPrompt exploits a structured-visual VKG pathway that is both distinct from and stronger than existing typographic jailbreaks.

---

> ### Author Response · Authors · 2025-11-20
>
> ### **W3: Lack of experimental defense evaluation**
>
> >_The paper mentions potential defenses (structure-aware filtering, uncertainty-aware refusal) but does not experimentally evaluate any defense mechanisms; this weakens the impact of the attack paper._
>
> We thank the reviewer for highlighting this point. In the revised manuscript, we therefore **implement and evaluate a potential defense against GraphPrompt**, as described in Sec.4.5 and Appendix C.
>
> Specifically, we add an **intent-first system prompt** that requires the target MLLM to (i) first **summarize the intent of the VKG image in natural language**, and (ii) then **decide whether this intent is policy-compliant before providing any substantive answer**. In effect, the model is explicitly asked to understand “what the graph is trying to do” and perform a safety check before continuing along the parse–then–execute pathway.
>
> We evaluate this defense on the same GraphPrompt-generated VKGs across all six target MLLMs. As shown in Appendix C, this intent-first defense **consistently reduces ASR for all models**, sometimes by a noticeable margin, but **does not eliminate** GraphPrompt’s attacks—non-trivial ASR remains even under this stricter setting. This indicates that such a prompt-based defense is **partially effective**, but insufficient to fully neutralize VKG-based jailbreaks.
>
> We now also emphasize in the conclusion that our defense experiment should be viewed as an initial step, and that designing more powerful **structure-aware defenses** for structured visual inputs (e.g., VKG-specific filters, cross-modal consistency checks, or uncertainty-aware refusal conditioned on graph structure) is an important direction for future work building on GraphPrompt.
>
>
>
> **Table 5:** Effect of adding a system-level safety defender on attack success rate (ASR, %).
> Each entry is computed over 20 harmful queries per model.
> The last row reports ΔASR (defender − no defender, in percentage points).
>
> | Prompt type            | GPT-4o | GPT-5mini | GPT-5 | Qwen 2.5 | Claude | Gemini |
> |------------------------|:------:|:---------:|:-----:|:--------:|:------:|:------:|
> | No defender (ASR)      |   90   |    85     |  95   |   100    |   80   |  100   |
> | +System defender (ASR) |   65   |    75     |  70   |    95    |   60   |   95   |
> | ΔASR (pp)              |  -25   |    -10    | -25   |    -5    |  -20   |   -5   |

---

> ### Author Response · Authors · 2025-11-20
>
> Dear Reviewer JRvZ,
>
> Again, we appreciate Reviewer JRvZ’s careful review and constructive criticisms. We hope that our clarifications on dataset scope, the structured-visual nature and mechanistic distinctiveness of GraphPrompt relative to FigStep, and our initial experimental defense evaluation help address your concerns.
>
>
> Best regards,
>
> The Authors

---

> > ### Comment · Reviewer_JRvZ · 2025-11-26
> >
> > Thank you for your detailed response. This paper is solid and experiments are comprehensive. I'll maintain my weak reject recommendation since the contribution of this work feels incremental to me. Although it's good to have a new jailbreaking method, it's already known that VLLMs are susceptible to jailbreak attacks, so it does not significantly advance our understanding about VLLM and VLLM safety.

---

> > > ### Author Response · Authors · 2025-11-26
> > >
> > > Dear Reviewer JRvZ,
> > >
> > > Thank you again for the careful reading and for clearly articulating your concerns.
> > >
> > > We fully agree that prior work has already established that VLLMs are vulnerable to jailbreak attacks in general. Our intent in this paper is not to claim “jailbreaks are new”, but rather to make progress on **where the remaining blind spots are and why existing safety mechanisms fail there**. Concretely, we see our contribution to understanding VLLM safety along three axes:
> > >
> > > 1. **A previously under-examined attack surface: structured visual inputs (VKGs).**
> > >    Existing work on multimodal jailbreaks has primarily focused on natural images or typographic overlays. In contrast, our results show that *visual knowledge graphs*—which are increasingly becoming first-class inputs in data analytics and tool-augmented systems—induce a qualitatively different *parse-then-execute* reasoning pathway that is not well covered by current safety alignment. To the best of our knowledge, this specific modality and pathway had not been systematically studied.
> > >
> > > 2. **A standardized VKG-based evaluation pipeline, not just one-off prompts.**
> > >    Instead of hand-crafted examples, GraphPrompt-Synth + GraphPrompt-Eval provide a reusable pipeline that (i) automatically generates high-quality adversarial VKGs from textual queries and (ii) yields a standardized benchmark for structure-aware safety evaluation across models. We believe this is useful for the community as a diagnostic tool, similar in spirit to how FigStep and MM-SafetyBench have shaped the evaluation of other visual attack surfaces.
> > >
> > > 3. **Mechanistic evidence on how structured vision bypasses safety layers.**
> > >    Beyond ASR numbers, we use a HiddenDetect-style analysis to show that VKG inputs systematically suppress safety-related activations in the identified “safety layers”, much more than text-only prompts or other visual baselines. Together with our ablations on structural complexity and resolution, this suggests that the vulnerability is tied to the model’s structured reasoning pathway rather than just another surface-level prompt variation.
> > >
> > > We understand that, from one perspective, this can still be viewed as “another jailbreak method”. Our goal, however, is to help map out and characterize a specific, increasingly relevant modality (VKGs) where current VLLM safety does not behave as expected, and to provide tools and evidence that future work on structure-aware defenses can build upon. We appreciate your feedback and hope that this clarification helps articulate the intended contribution of our work.
> > >
> > > Best regards,
> > >
> > > The Authors

---

### Official Review · Reviewer_XpE2 · 2025-11-01

**Soundness:** 2
**Presentation:** 2
**Contribution:** 3
**Rating:** 2
**Confidence:** 3

**Summary:**

This paper introduces GraphPrompt, a novel black-box jailbreaking framework that exploits the structural and semantic properties of Visual Knowledge Graphs (VKGs) to bypass safety alignments in Multimodal Large Language Models (MLLMs).

**Strengths:**

1. Novelty: This is the first work to systematically explore the security risks posed by VKGs in MLLMs, leveraging structural and semantic paradoxes for adversarial attacks.
2. Clarity and Coherence: The paper is well-structured and written with reasonable experimental design.
3. Practical Relevance: The attack framework is practical and poses a realistic threat to deployed MLLMs.

**Weaknesses:**

Methodological Detail Lacked: The paper lacks sufficient detail in key parts of the method. For example:
1.How exactly is semantic decomposition and topology-borne encoding performed?
2.How are visual encoding parameters (e.g., color, layout) adjusted during optimization?
3.How are graph size parameters (|V|, |E|) controlled or modified?

Dataset Scale: The use of SafeBench-Tiny (only 50 queries) limits the statistical reliability and generalizability of the results.

VKG Generation Process: It is unclear which model or tool is used to generate VKGs from Mermaid code, and how the quality or diversity of generated graphs is ensured.

Judge Model Validation: Although manual spot-checking is mentioned, there is no quantitative evaluation of the judge model’s accuracy or consistency.

Inconsistent Model Usage: Not all six models from Table 1 are included in the following experiments, which limits the completeness of the analysis.

**Questions:**

1. How was the semantic decomposition step implemented? Was it rule-based or model-based?
2. What was the rationale behind the ongoing contest scenario in the user prompt? How might this influence model behavior?
3. Why were only some of the six models used in the ablation studies?
4. Was the judge model’s performance evaluated? If so, what were the results?
5. See Weaknesses for more questions.

---

> ### Author Response · Authors · 2025-11-20
>
> We sincerely thank Reviewer XpE2 for the careful evaluation of our work, and we appreciate your recognition of the novelty, coherence, and practical relevance of GraphPrompt. Your comments on methodological detail, dataset scale, VKG generation, judge validation, and model coverage are very helpful. Below we outline how we clarify and extend the paper in response to your concerns.
>
>
>
> ### **W1: Methodological details of GraphPrompt**
>
> >*The paper lacks sufficient detail in key parts of the method, including:*
>
> 1. *How semantic decomposition and topology-borne encoding are performed;*
> 2. *How visual encoding parameters (e.g., color, layout) are adjusted during optimization;*
> 3. *How graph size parameters ($|V|$, $|E|$) are controlled or modified.*
>
> Thank you for raising this point. We agree these implementation details should be clearer, and we have made them explicit in Sec. 3.2–3.3 and Appendix A.2 of the revised manuscript.
>
> **Semantic decomposition & topology-borne encoding.**
> After obtaining the rewritten benign-surface query ($q_{\text{rew}}$), we perform semantic decomposition via a prompted LLM. Concretely, we use a graph-builder LLM (DeepSeek-R1) with the VKG-generation prompt in Appendix A.2 (Figure 5), which asks the model to (i) identify key entities, sub-tasks, and dependencies in ($q_{\text{rew}}$), (ii) group them into modules (e.g., “problem”, “preparation”, “process”, “analysis”), and (iii) output a Mermaid VKG with these elements as nodes and edges. This is what we term topology-borne encoding: the harmful process is distributed across the graph’s workflow-like structure rather than expressed as a single explicit instruction.
>
> **Visual encoding parameters (color, layout).**
> In the iterative optimization loop, we do **not** tune visual rendering parameters. Resolution, color theme, background, and layout engine are fixed by the configuration of our local Mermaid-based renderer (a static config). The optimization acts only on graph content and topology. Visual factors (e.g., color schemes, background styles) are varied **only** in separate, controlled ablation experiments, not as part of GraphPrompt-Synth.
>
> **Graph size control ($|V|$, $|E|$).**
> During refinement, we choose an appropriate optimization prompt (“simplify” or “enrich”) based on the previous round’s judge feedback, using the JSON-style prompts in Appendix A.2 (Figure 6). These prompts explicitly ask the LLM to adjust the graph’s topology and scale—splitting or merging nodes under the “problem” node, with target ranges on the number of nodes. After each refinement, we parse the Mermaid code to measure ($|V|$) and ($|E|$); if the graph is still outside the desired range, we continue refinement up to a fixed maximum number of iterations.

---

> ### Author Response · Authors · 2025-11-20
>
> ### **W2: Dataset scale (SafeBench-Tiny, 50 queries)**
> >*Using SafeBench-Tiny (only 50 queries) may limit statistical reliability and generalizability.*
>
> We agree that relying on SafeBench-Tiny means the number of harmful queries is relatively small, and this is a limitation of our current study. Our choice was driven by two considerations:
>
> Our choice of SafeBench-Tiny was motivated by two factors:
> (1) it is a **diverse** set of high-risk harmful queries that is already used in prior safety work, which allows **direct comparison** against existing multimodal jailbreak baselines such as FigStep under a shared benchmark; and
> (2) each harmful query is non-trivial, and we evaluate every query across six strong MLLMs with multiple attack variants and a GPT-5 judge, plus ablation settings, so we must control the overall experimental cost, which is dominated by repeated queries to the target models and the judge rather than by the number of distinct harmful queries.
>
> Even under this modest query count, the effect sizes are large and consistent: GraphPrompt achieves substantially higher ASR than MM-SafetyBench and FigStep on all six models, with aligned trends across query categories and in our ablation studies. This suggests that our main claims—namely, that VKG-based attacks open a new structured-visual attack surface and that GraphPrompt reliably outperforms existing multimodal jailbreaks—do not hinge on a single outlier subset of harmful queries.

---

> ### Author Response · Authors · 2025-11-20
>
> ### **W3: VKG generation from Mermaid code**
>
> >_It is unclear which model/tool is used to generate VKGs from Mermaid code, and how the quality/diversity of graphs is ensured._
>
>
> Thank you for pointing this out. We have clarified the VKG generation toolchain and our quality/diversity checks in the revised manuscript (Sec. 3.3 and Appendix A.2).
>
> - **Toolchain (LLM → Mermaid → VKG image).**
>     In our pipeline, the _structure_ of the VKG is produced by a **graph-builder LLM** (DeepSeek-R1), which outputs **Mermaid code** using the prompts in Appendix A.2 (Figure 5 / Figure 6). The Mermaid code is then rendered into VKG images using a **local Mermaid CLI renderer** with a fixed configuration (layout engine, resolution, background, font). Thus:
>     - LLM = DeepSeek-R1 (semantic decomposition + graph construction in Mermaid);
>     - Rendering tool = Mermaid CLI (Mermaid → PNG VKG image).
>         We state this explicitly in Sec. 3.3 to avoid any ambiguity.
> - **Quality control.**
>     We enforce quality at two levels:
>
>     1. **Syntactic/rendering validity:** after each LLM step we check that the output is valid Mermaid; code that fails to parse or render (e.g., syntax errors, empty graphs) is rejected, and the refinement loop is re-run until a valid VKG is produced or the iteration budget is exhausted.
>
>     2. **Structural adequacy:** the simplify/enrich prompts (Appendix A.2, Figure 6) explicitly require non-trivial structure (e.g., minimum node counts, multiple modules under the “problem” node).
>
> - **Diversity.**
>     Diversity arises from both the **input space** and the **generation process**:
>     - SafeBench-Tiny provides harmful queries from multiple categories, which naturally induce different workflows.
>     - For each query, the LLM-based construction and refinement (simplify/enrich) produces **varied topologies** (different decompositions into sub-nodes, branching patterns, and module structures), subject to node/edge constraints.

---

> ### Author Response · Authors · 2025-11-20
>
> ### **W4: Judge model validation**
>
> >_Manual spot-checking is mentioned, but there is no quantitative evaluation of the judge model’s accuracy or consistency._
>
> Thank you for pointing this out. We have now made the judge evaluation explicit and quantitative in the revised manuscript (Appendix B.2, Tables 7 and 8).
>
> Concretely, we evaluate the GPT-5 judge on a held-out subset of query–response pairs, comparing its ternary labels
> {explicit_refusal,violates_policies,answers_original_question}
> against human annotations for all six target models. As reported in Tables 7 and 8 of Appendix B.2, the AI / Manual rates for each dimension (e.g., “96 / 96”, “96 / 92”, “98 / 98”) are either identical or differ by only a few percentage points, with no systematic bias across models or label types.
>
> Moreover, our ASR is defined **jointly** over these three dimensions (no explicit refusal, does violate policies, and does answer the original question). This multi-criteria definition makes the final ASR label robust: even if there are small discrepancies on an individual dimension, their impact on the aggregate jailbreak decision is limited.
>
>
> **Table 7:** Comparison of AI-based vs. manual evaluation across six target models on three safety and utility dimensions.
> Entries are AI / Manual rates (%).
>
> | Dimension                | GPT-4o | GPT-5mini | GPT-5 | Qwen 2.5 | Claude | Gemini |
> |--------------------------|:------:|:---------:|:-----:|:--------:|:------:|:------:|
> | Explicit refusal         |  0/0   |   6/0     |  2/0  |   0/0    |  0/0   |  0/0   |
> | Harmful / violates policies | 96/96 | 96/92  | 98/98 | 100/98   | 84/82  | 100/100 |
> | Answers original question| 96/96  | 98/92     |100/98 |  98/98   | 80/80  | 100/100 |
> | Jailbreak success (ASR)  | 96/96  | 94/92     |98/98  |  98/98   | 80/80  | 100/100 |
>
>
>
>
>
> **Table 8:** Accuracy of GPT-5 as an automatic ASR judge compared to manual evaluation (%).
>
> | Judge type | GPT-4o | GPT-5mini | GPT-5 | Qwen 2.5 | Claude | Gemini | Overall |
> |------------|:------:|:---------:|:-----:|:--------:|:------:|:------:|:-------:|
> | GPT-5      |  100   |    98     |  100  |   100    |  100   |  100   |  99.7   |

---

> ### Author Response · Authors · 2025-11-20
>
> ### **W5: Inconsistent model usage across experiments**
>
> >_Not all six models from Table 1 are used in the following experiments (e.g., ablations), limiting the completeness of the analysis._
>
> Thank you for highlighting this issue. Our original intent was to (i) evaluate **all six models** in the main attack comparison, and (ii) use a **representative subset** of models for the more expensive ablations, mainly for cost and runtime reasons. We agree that this design choice was not clearly communicated and could leave the analysis feeling incomplete.
>
> In the revised manuscript, we have **extended all ablation studies to the remaining models** and updated the corresponding tables and discussion.
>
> These extended results confirm that our main conclusions about the sensitivity of VKG-based jailbreaks to structural and visual factors are **robust across all six MLLMs**, not just the subset originally shown. We also clarify in the ablation section that all six models are now included, so the experimental coverage is explicit.

---

> ### Author Response · Authors · 2025-11-20
>
> Dear Reviewer XpE2,
>
> Again, we are grateful for your detailed and constructive review. We believe that the additional methodological clarifications, judge validation summary, and explicit discussion of dataset scale and model coverage in the revised manuscript will address your concerns and better convey the scope and implications of GraphPrompt.
>
> Best regards,
>
> The Authors

---

> > ### Comment · Reviewer_XpE2 · 2025-11-26
> >
> > Thank you for your detailed and thoughtful responses to my review. I appreciate the effort you have made. However, I still have a few remaining concerns and questions:
> >
> > 1. Model Choice for Graph Construction: Except DeepSeek-R1 as the graph-builder LLM, have you experimented with other LLMs (e.g., GPT-4, Claude, or open-source alternatives) to assess the generalizability or performance variability of the graph construction process? This could be an important factor for reproducibility and robustness.
> > 2. Determination of the "Desired Range" for |V| and |E|: While its clarified node and edge counts are controlled via simplify/enrich prompts, it remains unclear how the “desired range” for |V| and |E| is initially defined or dynamically updated during optimization. Could you provide more details on the criteria or heuristics used to set these ranges?
> > 3. Dataset Scale: I still believe that 50 queries may be insufficient to fully capture the variability and real-world robustness of the attack.
> > 4. Human Evaluation Details: In the judge validation, strong agreement between GPT-5 and human annotators are reported. Could you specify how many human annotators were involved in this process and how their opinions differed?
> > 5. Model Selection Across Experiments: Noticed that not all six models from Table 1 were included in the ablation studies (e.g., GPT-4o and Gemini are missing in Tables 2–4). Could you explain the rationale for selecting different subsets of models across experiments?

---

> > > ### Author Response · Authors · 2025-11-26
> > >
> > > **We thank the reviewer for the constructive follow-up questions. Below we provide point-by-point clarifications.**
> > >
> > >
> > > ### **Q1. Model choice for graph construction**
> > > >_The paper only evaluates DeepSeek-R1 as the graph-builder. Are other LLMs tested to assess generalizability?_
> > >
> > > Thank you for raising this point—we agree that the choice of graph-builder LLM is important for both reproducibility and robustness.
> > > 1. **Pilot comparison with alternative builders.**
> > >     Before finalizing DeepSeek-R1 as GB, we conducted small-scale pilot experiments on a subset of harmful queries using:
> > >     - **GPT-5** as the graph-builder: we observed that, due to its strong built-in safety alignment, GPT-5 frequently injected **additional “educational” or “warning” nodes** into the graph (e.g., legal disclaimers, safety reminders). These nodes altered the intended harmful workflow and partially neutralized the attack semantics, which is not the behavior we wanted to study in this work.
> > >     - **GPT-4o and the open-source DeepSeek-V3**: in contrast, these models sometimes **failed to fully follow our graph-construction prompt** or the rewritten query, leading to VKGs that **omitted key steps or did not accurately preserve the intended entity–relation topology**. Such graphs were less faithful encodings of the underlying malicious workflow and reduced the reliability of the attack generation process.
> > > 2. **Rationale for choosing DeepSeek-R1.**
> > >     Empirically, **DeepSeek-R1** provided the most stable behavior among the tested options:
> > >     - it reliably adhered to our prompt constraints (e.g., 30–50 nodes, explicit control-flow structure),
> > >     - it accurately captured the multi-step topology implied by the rewritten query, and
> > >     - it **did not introduce extra safety/educational nodes** that would confound the attack semantics.
> > >         This made DeepSeek-R1 a practical and consistent choice for instantiating GB in our main experiments.
> > >
> > >
> > >
> > > ### **Q2. Determining the “desired range’’ of |V| and |E|**
> > > >_How is the initial / dynamic desired range of nodes and edges defined or updated during optimization?_
> > >
> > > Thank you for pointing out that our description of the node/edge heuristics was not sufficiently precise. We clarify below how the ranges were chosen and used.
> > >
> > > 1. **Initial node range (|V|) is fixed and comes from pilot studies.**
> > >     Before large-scale generation, we ran pilots across different target models (including GPT-5, Claude-Sonnet-4, Gemini-2.5-Flash, etc.) while varying node counts. We observed that:
> > >     - graphs with **<20 nodes** were often too sparse to encode the full multi-step malicious workflow (lower ASR),
> > >     - graphs with **>50–60 nodes** tended to be visually cluttered and less reliably parsed.
> > >         We therefore fixed a **global node budget of 30–50 nodes** for all models, and encoded this directly in the initial graph-builder template (“Create 30–50 nodes in the graph”). The baseline graphs used in Table 3 are drawn from this setting and have **≈40 nodes on average**.
> > > 2. **Dynamic adjustment in optimization uses small, bounded steps.**
> > >     In the optimization loop (Algorithm 1), the `simplify_graph` / `enrich_graph` templates do not introduce a new notion of “desired range”; instead, they **nudge** the node count within nearby bands to refine structure:
> > >     - `simplify_graph` gradually tightens the range (30–40 → 30–35 → 25–30 nodes) to remove redundant structure while keeping a reasonably rich workflow.
> > >     - `enrich_graph` expands the range (e.g., 40–50, and in the final attempt “no fewer than 50 nodes”) to add intermediate steps when the graph is too simple.
> > >         In practice, each iteration changes (|V|) by roughly **5–10 nodes**, and most successful adversarial VKGs still lie near the **∼40-node** region reported in Table 3.
> > > 3. **Edges (|E|) are implicitly controlled by flow semantics.**
> > >     We do not explicitly constrain (|E|); edges arise from the Mermaid flowchart semantics, as each transition or dependency becomes an edge. Empirically this yields **moderate-density graphs** (about 1–2 outgoing edges per node).

---

> > > ### Author Response · Authors · 2025-11-26
> > >
> > > ### **Q5. Model selection across experiments**
> > > >_Why do Tables 2–4 appear to omit some models?_
> > >
> > > We apologize for the confusion. After double-checking the revised manuscript, we confirm that **Tables 2–4 already include all six target models** (GPT-4o, GPT-5mini, GPT-5, Qwen-2.5, Claude-Sonnet-4, Gemini-2.5-Flash), and all analyses have been updated accordingly. The omission was likely due to page breaks or formatting when viewing the PDF.
> > >
> > > To avoid any ambiguity due to PDF rendering or page breaks, we also reproduce the full tables in this response thread for the reviewer’s convenience.
> > >
> > > **Table 2: Ablation on VKG rendering styles (ΔASR in percentage points).**
> > > Baseline ASR (top row) and changes for each variant. *No color* removes colors from nodes/edges; *White background* uses `#FFFFFF`; *Dark-red background* uses `#8B0000`.
> > > | Variant                 | GPT-4o | GPT-5mini | GPT-5 | Qwen 2.5 | Claude | Gemini |
> > > |-------------------------|:------:|:---------:|:-----:|:--------:|:------:|:------:|
> > > | **Baseline (ASR, %)**   |  96    |    92     |  98   |    98    |   80   |  100   |
> > > | No color (nodes/edges)  |  −2    | **+6**    | **+2** | **+2**   | **+4** |   0    |
> > > | White background        |  −4    | **+4**    |   0   |    0     | **+2** |   0    |
> > > | Dark-red background     |  −2    |    0      |   0   | **+2**   | **+2** |   0    |
> > >
> > > **Table 3: Graph complexity ablation (ΔASR in pp relative to baseline ~40-node graphs).**
> > > Positive values indicate higher ASR; negative values indicate degradation.
> > > For the **≤5 nodes** condition, only **30 queries** were used (10 categories × 3 each) whose pruned graphs still preserved harmful intent.
> > > | Node cap                     | GPT-4o   | GPT-5mini | GPT-5    | Qwen 2.5 | Claude      | Gemini  |
> > > |------------------------------|:--------:|:---------:|:--------:|:--------:|:-----------:|:-------:|
> > > | **Baseline (~40 nodes, ASR %)** | 96.00    | 92.00     | 98.00    | 98.00    | 80.00       | 100.00  |
> > > | ≤ 20 nodes                   | 0.00     | 0.00      | −4.00    | 0.00     | **+16.00**  | −2.00   |
> > > | ≤ 10 nodes                   | −2.00    | −2.00     | −14.00   | −2.00    | **+20.00**  | −4.00   |
> > > | ≤ 5 nodes (n = 30)           | −49.33   | −45.33    | −74.00   | −44.67   | −30.00      | −50.00  |
> > >
> > > **Table 4: Resolution ablation (ΔASR in pp).**
> > > We vary the renderer’s linear scale factor *s* (width and height scale linearly with *s*, pixel area with *s²*).
> > > The top row reports baseline ASR (%) at `scale=2`; other rows report changes relative to this baseline (negative = lower ASR).
> > > | Resolution (scale)              | GPT-4o | GPT-5mini | GPT-5 | Qwen 2.5 | Claude | Gemini |
> > > |---------------------------------|:------:|:---------:|:-----:|:--------:|:------:|:------:|
> > > | **Baseline (ASR, `scale=2`)**   |  96    |    92     |  98   |    98    |   80   |  100   |
> > > | Quarter (`scale=0.5`)           | −24    |    −4     |  −8   |   −28    |  −30   |   −6   |
> > > | Very-low (`scale=0.3`)          | −60    |   −50     | −56   |   −62    |  −68   |  −44   |

---

> > > ### Author Response · Authors · 2025-11-26
> > >
> > > We again thank reviewer XpE2 for the insightful comments and hope that these clarifications help better convey the intended contribution of our work.
> > >
> > > Best regards,
> > >
> > > The Authors

---

> > > > ### Comment · Reviewer_XpE2 · 2025-11-27
> > > >
> > > > Thanks a lot for the thorough response. I appreciate the detailed clarifications. But considering the large revisions necessary to both the method and the experiment parts of the submitted draft, I still suggest this work to be resubmitted.
> > > >
> > > > One more concern is inconsistency in human evaluation sample size:
> > > >
> > > > You mention that the human evaluation was performed on a sample of 20 queries (resulting in 120 model outputs). However, the percentages reported in Tables 7 and 8 (e.g., 96%, 98%, 100%) suggest that the human evaluation may have been conducted on a larger sample—specifically, one that is a multiple of 50, as these percentages are all multiples of 2%. Could you please clarify whether the human evaluation was in fact conducted on 20 queries or on a larger set? If it was indeed 20 queries, please explain how the reported percentages (e.g., 96%, 98%) were derived from a sample of 120 outputs, as these values would require fractional counts that are unlikely to arise in practice.
> > > >
> > > > Besides, while I understand the cost constraints involved in black-box evaluation, the use of only 50 base queries may limit the statistical reliability and real-world generalizability of the findings. Although SafeBench-Tiny provides broad category coverage, the relatively small number of queries per category (only 5) increases the risk that results are influenced by a few specific prompts.

---

> ### Author Response · Authors · 2025-11-26
>
> ### **Q3. Dataset scale (50 queries)**
> >_50 queries may be insufficient to capture real-world variability._
>
> We appreciate this concern and agree that dataset scale is an important dimension of robustness. Our choice of 50 queries reflects a trade-off between **risk coverage** and **black-box evaluation cost**.
> 1. **Coverage and experimental breadth.**
>     - We adopt **SafeBench-Tiny**, which contains **50 harmful queries spanning 10 high-risk categories** (e.g., illegal activity, hate speech, malware, physical harm, fraud, adult content, privacy, legal/financial/health advice). This gives us broad coverage over safety-relevant behaviors despite the modest query count.
>     - Each query is instantiated into **three VKG variants** and evaluated across **six heterogeneous MLLMs** (GPT-4o, GPT-5mini, GPT-5, Qwen-2.5-VL-72B, Claude-Sonnet-4, Gemini-2.5-Flash), yielding a large number of VKG–model interactions in the main experiment, plus additional runs for the ablation studies and defense evaluation.
>     - Beyond ASR, we also conduct **mechanistic analysis** of internal refusal dynamics (Sec. 4.4, Fig. 9 / App. E), which consistently shows that VKG-based attacks suppress safety-related activations compared to text-only prompts and other visual baselines. Taken together, **these results suggest that the vulnerability we uncover is structural rather than an artifact of a few specific prompts.**
> 2. **Limitation and future scaling.**
>     We nevertheless acknowledge that relying on 50 base queries is a limitation of the current study and will state this more explicitly in the paper. A key motivation for **our plan to release** the code and VKG generation pipeline is to enable future work to **scale GraphPrompt-style evaluation to larger harmful corpora** and more diverse domains **when budget permits**.
>
>
> Thank you for asking for more detail on the human evaluation; we are happy to clarify.
> ### **Q4. Human evaluation details**
> > _How many annotators were involved? How large was the annotated sample? How often did they disagree?_
>
> Thank you for asking for more detail on the human evaluation; we are happy to clarify.
> 1. **Sample size and construction.**
>    We used **all 50 harmful queries** from SafeBench-Tiny . For each query, we collected responses from all **6 target models**, yielding **300 model outputs** for manual review.
> 2. **Annotators and protocol.**
>    These 300 outputs were evenly distributed among **6 researchers** with experience in ML and safety. Each annotator independently labeled their assigned outputs along the three dimensions \((r, v, a)\) using exactly the same rubric and instructions as our GPT-5 judge (Appendix B.1, Figure 7).
> 3. **Disagreements and consensus.**
>    After independent labeling, annotators discussed 8 responses they were uncertain about and resolved them to obtain a consensus label (8 out of 300 responses, ≈2.7\%). When we then compared GPT-5’s labels against these consensus labels, GPT-5 achieved ≥98\% accuracy per model and 99.7\% overall (Table 8), indicating that the automatic judge is sufficiently reliable for our large-scale ASR measurement.

---

> ### Author Response · Authors · 2025-11-27
>
> Thank you for the careful follow-up and for pointing out the inconsistency in our description of the human evaluation.
>
> You are correct that there is a mismatch: **Tables 7 and 8 are based on 50 samples per model (N = 50, 300 outputs in total)**, which is why all percentages are multiples of 2%. The earlier statement that “human evaluation was performed on 20 queries (120 outputs)” is our mistake: the 20-query subset is used only for the defense/prompt-sensitivity experiments, but we inadvertently reused that description when summarizing the judge-validation study. In the revised manuscript, we will (i) explicitly state N = 50 per model in Appendix B and in the captions of Tables 7 and 8, and (ii) remove or correct any references to “20 queries / 120 outputs” in this context. The underlying human annotations and **the reported percentages in Tables 7 and 8 are correct**; only the textual description of N was inaccurate.
>
> Regarding the use of only 50 harmful queries from SafeBench-Tiny, **we acknowledge this as a limitation driven by our evaluation budget and state it in the limitations paragraph of Sec. 6.** Under this shared 50-query benchmark, GraphPrompt not only consistently outperforms FigStep and other strong baselines across all six models and metrics, but—crucially—**our mechanistic analysis shows that VKG inputs suppress internal safety activations more strongly than FigStep**. This indicates a **structural vulnerability inherent to the VKG modality**, rather than an artifact driven by a few particular prompts.

---

> ### Author Response · Authors · 2025-11-27
>
> Dear Reviewer XpE2,
>
> We sincerely thank you for the thoughtful comments and constructive suggestions. Your feedback has provided valuable guidance for improving both the clarity and rigor of our work, and has helped us better identify key directions for strengthening the method, experiments, and overall presentation. We appreciate the time and effort you invested in reviewing our submission.
>
>
> Best regards,
>
> The Authors

---

### Official Review · Reviewer_3Na9 · 2025-11-01

**Soundness:** 3
**Presentation:** 3
**Contribution:** 3
**Rating:** 6
**Confidence:** 3

**Summary:**

This paper introduces GraphPrompt, a novel black-box jailbreaking framework that exploits Visual Knowledge Graphs (VKG) to bypass safety alignment in MLLMs. By embedding harmful intents into graph topologies and pairing them with benign textual prompts, GraphPrompt induces a "parse-then-execute" pathway that evades text-based filters. The work underscores critical vulnerabilities in MLLMs’ cross-modal reasoning and proposes future defenses centered on structure-aware safety mechanisms.

**Strengths:**

- Compelling experimental results: The paper achieves exceptionally high ASR under strict black-box settings, significantly outperforming strong baselines like MM-SafetyBench and FigStep.

- Rigorous ablation analysis: The systematic ablation studies (e.g., varying node count, resolution, color schemes) clearly illustrate how graph topology and visual encoding affect VKG-driven attack success

- Insights for future defense insights: The success of VKG attack inspires future efforts toward VKG-like jailbreaks.

**Weaknesses:**

- Mechanistic explanation of attack efficacy: While the paper empirically demonstrates high ASR, it lacks a deeper theoretical or mechanistic explanation of why VKG so effectively bypasses safety alignment. For instance, how does graph parsing alter the model’s internal reasoning trajectory?

- Cost analysis of black-box optimization: The feedback loop involves iterative querying, but the computational cost of generating adversarial VKGs is not quantified.

- Reference Issue: The manuscript mentions "Appendix 4" in Section 4.1, but no appendices are included in the submission.

**Questions:**

Please see the weakness part.

---

> ### Author Response · Authors · 2025-11-20
>
> We sincerely thank Reviewer 3Na9 for the careful evaluation of our work, and we are grateful for your positive assessment of our experimental results, ablation studies, and the broader impact of VKG-based jailbreaks on future defense research. In light of your helpful suggestions, we have conducted additional **analyses** and revised the manuscript accordingly. Below, we outline how we have addressed each of your concerns.
>
> ### **W1: Mechanistic explanation of attack efficacy**
> >_“While the paper empirically demonstrates high ASR, it lacks a deeper theoretical or mechanistic explanation of why VKG so effectively bypasses safety alignment. For instance, how does graph parsing alter the model’s internal reasoning trajectory?”_
>
> Thank you for raising this important point—we fully agree that going beyond ASR and understanding _why_ VKG-based attacks work is crucial. In the current manuscript, we already take a first step toward such a mechanistic explanation in Sec.4.4 and Appendix E by analyzing the internal behavior of Qwen-VL-Chat using a HiddenDetect-style approach.
> Concretely, for each transformer layer we learn a “refusal direction” from hidden states associated with refusal vs. non-refusal outputs, and define a layer-wise _refusal strength_ as the cosine similarity between the hidden state and this direction. We then compare four conditions on the same harmful prompts: (i) text-only harmful input, (ii) FigStep images, (iii) MM-SafetyBench images, and (iv) our VKG images from GraphPrompt. As visualized in Figure 9 of the revised manuscript, the resulting curves exhibit three consistent patterns:
> - Text-only harmful prompts **strongly activate** the safety-critical layers, indicating that explicit harmful wording robustly triggers the model’s internal safety signal.
> - All image-based attacks **suppress** refusal strength in these layers relative to text, suggesting that routing harmful content through the visual channel already weakens safety activation.
> - VKG inputs show the **lowest refusal strength** in the safety band among all four conditions, meaning that, when processing VKGs, the hidden representations in safety-critical layers are the least “refusal-like”.
>
> This directly supports our “parse-then-execute” hypothesis. A VKG recasts the harmful query as a structured workflow: nodes encode entities and steps, and edges encode control flow and dependencies. Our analysis indicates that under such inputs, the safety-critical layers are dominated by _procedural parsing and execution_ of the depicted workflow rather than by detecting overt harmful intent, which explains why VKG-based attacks can bypass safety alignment more effectively and achieve higher ASR.
>
> In the revision, we will make this connection more explicit by (i) clearly stating in Sec. 4.4 that the HiddenDetect analysis is designed to probe how VKG parsing alters the internal reasoning trajectory, and (ii) adding a brief takeaway that summarizes the three patterns above and crystallizes the mechanistic insight: VKGs are effective because they steer safety-critical layers away from refusal directions while preserving detailed procedural semantics of the harmful task.

---

> ### Author Response · Authors · 2025-11-20
>
> ### **W2: Cost analysis of black-box optimization**
> >_“The feedback loop involves iterative querying, but the computational cost of generating adversarial VKGs is not quantified.”_
>
> We appreciate this observation and agree that understanding the computational/monetary cost of GraphPrompt is important for assessing its practical impact.
>
> In the revised manuscript, we now **explicitly quantify the end-to-end cost** of generating adversarial VKGs under contemporary API pricing (Appendix D, Table 10). For each VKG, we log token usage across all stages of the pipeline—(i) graph initialization and refinement with DeepSeek-R1, (ii) black-box evaluation against three target validators (GPT-5 / GPT-4o / Qwen2.5-VL-72B), and (iii) judge-side evaluation with GPT-5—and convert these to USD using official pricing. The results show that:
> - The **average total cost per VKG** is **$0.0708**, with a **minimum of $0.0077** and a **worst case of $0.2313** per VKG.
> - Decomposed by stage, the **average per-VKG costs** are:
>     - Graph init & refinement (DeepSeek-R1): **$0.0082**
>     - Black-box evaluation across validators (GPT-5 / GPT-4o / Qwen2.5-VL): **$0.0431**
>     - Judge evaluation (GPT-5 critic): **$0.0195**
> - Thus, **graph construction contributes only a small fraction of the budget**, while the majority of the cost comes from querying target models and the judge.
>
> When combined with our efficiency results in Figure 3—GraphPrompt succeeds with **≈1.25 attempts per query on average**, substantially fewer than baselines such as MM-SafetyBench and FigStep (1.8–2.4 attempts)—this cost profile indicates that high-success VKG attacks can be mounted at **modest cost**. In other words, the per-sample cost is low enough that GraphPrompt is practically usable in red-teaming workflows and routine safety evaluations, rather than being restricted to small proof-of-concept experiments.
>
> **Table 10:** Estimated per-VKG generation cost by stage (USD per VKG).
> Costs are amortized over three target validators and computed assuming DeepSeek-R1 for graph construction,
> GPT-5.1 / GPT-4o / Qwen2.5-VL-72B as validators, GPT-5.1 as judge, and a 4800×3200 rendering (≈1105 image tokens).
>
> | Stage                          | Model(s)                        |   Min   |   Max   |   Avg   |
> |--------------------------------|---------------------------------|--------:|--------:|--------:|
> | Graph init & refinement        | DeepSeek-R1                     | 0.0041  | 0.0123  | 0.0082  |
> | Black-box evaluation (validators) | GPT-5 / GPT-4o / Qwen2.5-VL  | 0.0003  | 0.1800  | 0.0431  |
> | Judge evaluation (GPT-5 critic)| GPT-5                           | 0.0033  | 0.0390  | 0.0195  |
> | **Total per VKG**              | —                               | **0.0077** | **0.2313** | **0.0708** |

---

> ### Author Response · Authors · 2025-11-20
>
> ### **W3: Reference / appendix issue in Section 4.1**
> >_“The manuscript mentions ‘Appendix 4’ in Section 4.1, but no appendices are included in the submission.”_
>
> We apologize for this confusion and thank you for pointing it out. The reference to “Appendix 4” was an oversight from an earlier draft and should not have appeared in the submitted version.
> In the revised manuscript, we have **corrected this reference** and **included the complete set of appendices**.

---

> ### Author Response · Authors · 2025-11-20
>
> Dear Reviewer 3Na9,
>
> Again, we sincerely appreciate your careful review and constructive suggestions. We hope that our added HiddenDetect-based analysis of Qwen-VL-Chat’s internal refusal dynamics, which clarifies how VKG inputs steer safety-critical layers along a parse-then-execute trajectory, together with the new quantitative cost analysis of our black-box optimization pipeline and the correction and completion of the appendix references and materials, adequately address your concerns and further strengthen the contribution of this work.
>
> Best regards,
>
> The Authors

---

### Note · Authors · 2026-01-05

**Comment:**

I have read and agree with the venue's withdrawal policy on behalf of myself and my co-authors.

**Withdrawal Confirmation:**

I have read and agree with the venue's withdrawal policy on behalf of myself and my co-authors.